# Recombinant Human Erythropoietin Effects on Well-Trained Athletes’ Endurance Performance: A Systematic Review

**DOI:** 10.3390/sports13030078

**Published:** 2025-03-06

**Authors:** Aitor Alberdi-Garciandia, Jordan Santos-Concejero

**Affiliations:** Department of Physical Education and Sport, University of the Basque Country UPV/EHU, 01007 Vitoria-Gasteiz, Spain; aalberdi086@ikasle.ehu.eus

**Keywords:** sports performance, doping, EPO, exercise

## Abstract

The use of recombinant human erythropoietin (rHuEPO) has been found to improve different cardiopulmonary-related variables that ultimately enhance endurance performance. The main goal of this systematic review was to analyze the hematological, physiological, and performance effects (both maximal and submaximal) of rHuEPO in well-trained endurance athletes. A literature search was conducted in three different databases (PubMed, Web of Science, and Scopus) on 20 January 2025; including studies published from 1 January 2010 to the search date. After analyzing 985 resultant articles and 5 records identified outside of the databases through citation tracking, 10 studies that met the inclusion criteria were included in the systematic review. We found that, regardless of the total dose of rHuEPO used, this substance improves the main hematological (total hemoglobin mass, hemoglobin concentration, and hematocrit) and physiological (maximal oxygen uptake and peak oxygen uptake) parameters, while the maximal performance-related parameters (mainly, maximal power output, and peak power output) also tend to increase. However, further research is needed to determine if rHuEPO can also improve submaximal parameters, which are also major determinants of performance in endurance sports.

## 1. Introduction

Erythropoietin (EPO) is the hematopoietic growth factor responsible for stimulating erythropoiesis in response to cellular hypoxia [1]. EPO is a glycoprotein hormone characterized by a protein skeleton of 165 amino acids with four carbohydrate chains, each containing two to four sialic acid residues [2]. Most EPO is produced in the peritubular interstitial cells of the renal cortex and medulla, with approximately 15% synthesized in the liver [3]. EPO plays a central role in regulating erythrocyte production by recruiting and differentiating erythroid progenitor cells in the bone marrow, supporting their maintenance and survival, and stimulating hemoglobin (Hb) synthesis [4]. In the 1980s, recombinant human erythropoietin (rHuEPO) was developed to address clinical conditions requiring this vital hormone, particularly anemia associated with chronic renal dysfunction [5].

Beyond its clinical applications, rHuEPO gained notoriety as a doping agent in endurance sports such as road cycling, swimming, middle- and long-distance running, and triathlon. These sports rely on complex physiological determinants of performance, including maximal oxygen uptake (VO_2max_), effort economy, blood lactate concentration at various intensities ([La^−^]), and time to exhaustion (TTE) [6,7]. VO_2max_, the primary indicator of aerobic capacity, reflects the maximum oxygen volume a person can absorb, transport, and utilize during maximal exertion [8]. However, other parameters, such as effort economy, that is, the energy consumed per unit of velocity, emerged as critical predictors of endurance performance [9]. Additionally, power output in cycling and running times in middle- and long-distance events are key performance markers [10,11,12,13].

The primary goal of rHuEPO use in sports is to increase erythrocyte count, elevate Hb levels, and enhance oxygen delivery to tissues, thereby improving VO_2max_ and performance in aerobic exercise [14]. Despite its efficacy, rHuEPO belongs to the class of erythropoiesis-stimulating agents (ESAs) banned by the World Anti-Doping Agency (WADA) in 1990 [15,16].

Multiple reviews examined the impact of rHuEPO on endurance athletes. High-quality systematic reviews presented conflicting evidence: while some authors found no definitive performance-enhancing effects in elite cyclists despite increases in Hb, hematocrit (Hct), and VO_2max_ [17], others concluded that rHuEPO could enhance hematological and pulmonary parameters, maximal power output (P_max_), and TTE [18]. These improvements were most evident during maximal intensities, which could be less relevant for cyclic endurance sports that require sustained submaximal efforts.

Conversely, narrative reviews with less rigorous methodologies highlighted significant enhancements in hematological, pulmonary, and performance parameters [19,20]. However, the lack of systematic approaches in these studies raises concerns about selection bias.

Nonetheless, rHuEPO remains a controversial substance in sports, both for its ethical implications and potential health risks. Considering the scientific evidence published to date, this systematic review aims to investigate the hematological, physiological, and performance effects (both maximal and submaximal) of rHuEPO use in well-trained endurance athletes. We hypothesize that rHuEPO will yield clear improvements in these parameters in this population.

## 2. Materials and Methods

### 2.1. Search Strategy

A comprehensive literature search was conducted on 20 January 2025, across three major scientific databases (PubMed, Web of Science, and Scopus) to identify relevant studies. The review included studies published between 1 January 2010 and 20 January 2025. The procedure adhered strictly to the Preferred Reporting Items for Systematic Reviews and Meta-Analyses (PRISMA) guidelines to ensure transparency and reproducibility. Further details, including the PRISMA 2020 checklists for the abstract (Appendix A) and the main text (Appendix A), can be found in the Appendix A. The search in each database was performed in the areas of title, abstract, and keywords. For this purpose, combined with Boolean operators, different lines composed of keywords referring to different areas of the subject were used. Keywords in the same line were combined with the “OR” operator, while those in different lines were combined with the “AND” operator (Table 1).

Both authors (A.A.-G. and J.S.-C.) independently conducted a blinded search of study titles, abstracts, and keywords to ensure the accuracy and reproducibility of the results. After completing the initial selection, both researchers met to finalize the list of included studies. Subsequently, they independently conducted a full-text review of the eligible studies and reconvened to confirm the final selection. No discrepancies or ambiguities arose during the selection process, as both researchers reached a high level of agreement, unanimously confirming that all selected studies met the inclusion criteria.

Thus, the following combination of keywords and Boolean operators was used: (Erythropoietin OR EPO) AND (performance OR endurance) AND (athletes). Apart from the established date limits (1 January 2010–25 January 2025), no other limits or filters were used in the database searches.

### 2.2. Data Collection and Analysis

The results from the literature search were uploaded to Rayyan software (Rayyan Systems Inc., Doha, Qatar, web version), a powerful online tool designed to facilitate the methodology process in systematic reviews and meta-analyses [21]. The application allows us to analyze the abstract (introduction, objectives, methodology, results, and conclusions) of the general characteristics (type of publication, language, authors, etc.) of the studies found in the literature search. In addition, it offers the possibility of analyzing and classifying the studies one by one, being a valuable tool for carrying out systematic reviews [22].

### 2.3. Inclusion and Exclusion Criteria

Studies included in this systematic review met the following inclusion criteria: (a) studies with an experimental design (meta-analyses, systematic reviews, and other types of reviews are excluded); (b) studies conducted with human subjects; (c) studies in which participants were healthy, without any type of disease; (d) studies in which participants were endurance trained (VO_2max_ = ≥45 mL·kg^−1^·min^−1^ or ≥3.5 L·min^−1^, a threshold established based on the limits defined in previous study [23]); (e) studies in which at least one performance-related parameter was reported; and (f) studies in which at least one of the performance-related parameters can be objectively measured. After applying the inclusion and exclusion criteria, the following data were extracted from each study: name of the first author, year of publication, intervention and placebo group characteristics, rHuEPO dose, duration of intervention, measured key variables, and rHuEPO administration effects on measured key variables.

### 2.4. Assessment of Methodological Quality

To assess the methodological quality of the studies included in this systematic review, the Oxford Level of Evidence scale [24] and the Physiotherapy Evidence Database (PEDro) scale were used [25]. The Oxford Level of Evidence scale measures the level of evidence of a study on a scale ranging from level 1a to level 5, with level 1a being systematic reviews of high-quality randomized controlled trials and level 5 being expert opinion. On the other hand, the PEDro scale is made up of 11 different items related to scientific precision [26]. Both authors (A.A.-G. and J.S.-C.) independently conducted the quality assessments of the selected studies, and consensus on the scores was achieved through meetings.

## 3. Results

### 3.1. Study Selection

A total of ten papers were identified for inclusion in the review. Figure 1 presents detailed information on the study selection process.

The search of the electronic databases provided a total of 985 citations. After duplication removal, the final number of citations was 362. Of these, 345 were eliminated after screening the titles, abstracts, and keywords. Seventeen full-text papers were examined for final confirmation of eligibility criteria. Additionally, five records were identified outside of the databases through citation tracking. In total, twelve studies did not meet the inclusion criteria.

### 3.2. Methodological Quality and Level of Evidence

Table 2 presents the evaluation of the methodological quality of the studies using the PEDro scale (also used to evaluate the risk of bias for each study) and the Oxford Level of Evidence framework. The average PEDro score was 7.2, reflecting high methodological standards overall. Six studies were categorized as level 1b, indicating high-quality randomized controlled trials, while four were classified as level 2b, representing cohort studies or lower-quality trials. Key strengths included robust randomization, allocation concealment, and blinding of outcome assessors. However, limitations such as incomplete blinding of therapists and variability in reporting key outcomes were noted in some studies. Despite these issues, the strong design and high evidence level of most studies ensure robustness in outcome measurement. Nonetheless, caution is warranted due to heterogeneity in study designs and intervention protocols, which may influence the generalizability of the findings.

### 3.3. Tools and Protocols

To collect and analyze hematological variables, all studies included in this systematic review [10,11,12,13,27,28,29,30,31,32], employed blood gas analyzers as the primary measurement tool. These devices ensured accurate assessment of hematological parameters critical to evaluating the effects of rHuEPO administration.

In all the aforementioned studies, physiological variables were measured using a combination of tools and protocols. Specifically, a cycloergometer paired with a gas analyzer was used in conjunction with an incremental exercise protocol performed until exhaustion. This setup provided precise data on parameters such as oxygen consumption and ventilatory thresholds.

For studies focusing on maximal performance-related variables [11,12,13,28,30,31,32], a cycloergometer equipped with a power meter was employed. These measurements also relied on an incremental exercise protocol to exhaustion, ensuring reliable quantification of power output and related variables.

Finally, submaximal performance-related variables were assessed in several studies [10,11,12,13,27,30,31]. In this context, a cycloergometer or a bicycle equipped with a power meter was used, but the protocol differed. Instead of an incremental approach, a time trial protocol at submaximal intensity was implemented, allowing for the evaluation of sustained performance and efficiency under controlled conditions.

### 3.4. Study Characteristics

Table 3 indicates the characteristics of the included studies.

### 3.5. Key Findings

The tools and protocols employed to obtain results have been consistent across the studies included in this systematic review. The primary differences between studies lie in the dosage of rHuEPO administered and the duration of treatment. This consistency in methodology allows for meaningful comparisons between studies and facilitates the identification of causal relationships in the observed outcomes.

Significant increases in hematological variables were reported in 9 out of the 10 studies analyzed [10,11,12,13,27,28,29,31,32]. Total hemoglobin mass (tHb) increased by 6.7% to 19.7%, Hb by 8.3% to 16.2%, and Hct by 2.6% to 18.8% relative to baseline values.

Similarly, physiological variables demonstrated significant increases in 9 out of 10 studies [10,11,12,13,27,28,29,31,32]. The reported improvements included increases in VO_2max_ ranging from 2.6% to 10%, and peak oxygen uptake (VO_2peak_) ranging from 4.2% to 6.2% relative to baseline values.

Of the seven studies that analyzed maximal performance-related variables, four reported significant improvements [11,12,13,31]. Specifically, P_max_ and peak power output (P_peak_) increased by 2.7% to 5.8% relative to baseline values. On the other hand, submaximal performance-related variables showed significant improvements in 3 out of 7 studies [10,11,13]. Notable increases were observed in the constant load test limit time (t_lim_) during time trials, ranging from 4.3% to 69.7%, and in the 3000 m race time, which improved by 4.6% to 5.7% relative to baseline values.

The systematic review reveals that the analyzed studies do not demonstrate a dose-dependent effect of rHuEPO on the investigated variables. Specifically, no greater effect, reflected as a higher percentage increase relative to baseline values, has been observed in studies administering higher doses of rHuEPO, nor has a diminished effect been evident with lower doses.

## 4. Discussion

### 4.1. Interpretation of Findings

This systematic review suggests that the administration of rHuEPO leads to significant increases in key hematological parameters, including tHb, Hb, and Hct. These effects are physiologically attributable to the synthetic stimulation of erythrocyte production induced by rHuEPO, a member of the ESAs group [33,34]. Similar increases in hematological parameters have been consistently reported in other reviews investigating the effects of rHuEPO on endurance-trained athletes [17,18]. These hematological enhancements facilitate greater oxygen delivery to tissues [35], which, as observed in this review, translates into improvements in key physiological variables, particularly VO_2max_ and VO_2peak_. This aligns with findings from previous studies analyzing the same physiological outcomes in comparable populations and settings [17,20].

The importance of tHb as a key determinant of endurance performance has been further supported by recent findings [36], which establish reference values for tHb in well-trained endurance athletes. This study reports an average tHb of approximately 960 g in male athletes and 625 g in female athletes, aligning with the notion that endurance-trained individuals exhibit significantly higher values compared to non-athletes. These values are consistent with the increases observed in this systematic review, where rHuEPO administration led to significant enhancements in tHb across multiple studies. Given that tHb is more strongly correlated with VO_2max_ than Hb alone [37], these reference values provide a useful framework to contextualize the hematological effects of rHuEPO. Notably, the increases in tHb following rHuEPO administration in the studies analyzed in this review approach or even exceed the upper range of natural variations reported in endurance athletes [36]. Consequently, these findings highlight the necessity of advanced monitoring strategies that integrate individual baseline tHb values to improve the detection of artificial hematological manipulations.

In addition to hematological and physiological parameters, maximal performance-related variables, such as P_max_ and P_peak_, showed significant improvements in four out of the seven studies included in this review [11,12,13,31]. This indicates that increases in hematological and physiological variables can contribute to enhanced performance in over half of the studies, a finding corroborated by other reviews [20]. Conversely, the evidence for the effects of rHuEPO on submaximal performance variables, such as t_lim_ and running times over specific distances, remains inconsistent. While significant improvements were observed in only three out of seven studies in this review [10,11,13], previous reviews reported both positive changes [20] and negligible changes [17,18].

The findings of this systematic review suggest that rHuEPO administration, regardless of the total dose, consistently enhances key hematological and physiological parameters and tends to improve maximal performance-related outcomes. However, further research is required to elucidate its effects on submaximal parameters, which are critical determinants of performance in cyclic endurance sports.

The ergogenic effects of rHuEPO, particularly its capacity to enhance hematological and physiological parameters, pose a significant challenge for anti-doping detection methods [38,39]. This difficulty arises because, while rHuEPO administration leads to measurable increases in tHb, Hb, Hct, VO_2max_, and VO_2peak_, its pharmacokinetics, including a variable elimination half-life influenced by renal clearance and dosing strategies, can complicate detection [40]. Additionally, the use of microdoses, as suggested by recent research [41], may further obscure detection by sustaining performance-enhancing effects while keeping hematological fluctuations within the individual reference ranges of the athlete biological passport [42]. Furthermore, the structural similarity between exogenous rHuEPO and endogenous EPO necessitates highly sensitive analytical techniques, such as isoelectric focusing and mass spectrometry, to differentiate their isoforms [43]. Consequently, the interplay between the physiological enhancements of rHuEPO and current anti-doping limitations underscores the need for continued advancements in detection methodologies.

Considering all this, some competitive athletes continue to exploit the ergogenic effects of rHuEPO, disregarding its ban by WADA since the 1990s [15], and exposing themselves to severe health risks, including thrombotic complications [17].

### 4.2. Limitations

The present review has certain limitations that must be acknowledged. Firstly, the treatment durations and total doses of rHuEPO administered varied across most studies, with only two investigations utilizing the same protocol [10,12]. This heterogeneity complicates direct comparisons of the analyzed variables under identical dosing conditions. However, the diversity in dosing protocols allows for broader conclusions on the effects of varying rHuEPO doses. Notably, as observed in this review, the ergogenic effects of rHuEPO appear to be independent of the total dose, at least within the range examined in the included studies.

Secondly, the sample sizes in 6 of the 10 studies included in this review were relatively small (ranging from 6 to 29 participants) [10,27,28,29,30,32]. In contrast, the remaining four studies had larger sample sizes (ranging from 39 to 48 participants) [11,12,13,31]. This discrepancy in sample sizes introduces variability that poses challenges when making meaningful comparisons across studies. Additionally, only 3 of 10 studies included female participants [13,28,30], limiting the ability to draw sex-specific conclusions and underscoring the need for more inclusive research designs in this area.

### 4.3. Future Directions

Future research should focus on investigating the effects of rHuEPO on submaximal performance variables in cyclic endurance sports and well-trained athletes. These variables are crucial determinants of performance in this type of sport, yet the current evidence is insufficient to draw clear conclusions regarding their responsiveness to rHuEPO administration.

Furthermore, studies dedicated exclusively to analyzing the effects of microdoses of rHuEPO in endurance athletes are warranted. Microdoses may provide an ergogenic advantage [41], while evading detection by standard anti-doping protocols, such as the biological passport [42]. Developing robust methods to reliably detect microdose usage is essential to prevent athletes from gaining illicit advantages and to ensure fair competition.

## 5. Conclusions

This systematic review highlights the varying effects of rHuEPO on hematological, physiological, and performance parameters in well-trained athletes involved in endurance sports. Irrespective of the total dose administered, rHuEPO consistently enhances key hematological markers such as tHb, Hb, and Hct, alongside physiological parameters including VO_2max_ and VO_2peak_.

Regarding performance outcomes, rHuEPO administration tends to improve maximal performance variables, particularly P_max_ and P_peak_. However, evidence for its impact on submaximal performance parameters remains inconclusive, warranting further research to clarify its role in metrics that are critical determinants of endurance performance.

The findings underscore rHuEPO’s capacity to artificially augment oxygen transport and delivery by increasing erythrocyte mass, conferring a physiological advantage to athletes who utilize it illicitly. This reinforces the ethical and health concerns surrounding its use, particularly in competitions where such practices are prohibited, as it provides an unfair advantage and poses significant health risks.

## Figures and Tables

**Figure 1 sports-13-00078-f001:**
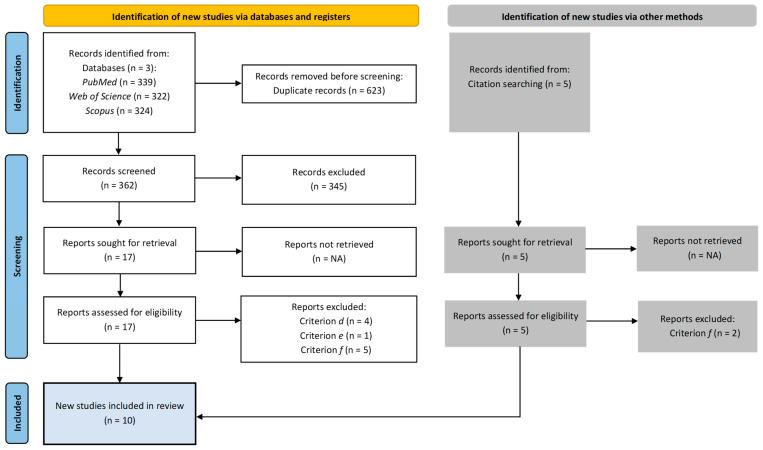
Study selection process.

**Table 1 sports-13-00078-t001:** Keywords and operators used in the search strategy.

Line	Keywords	Operators	
Reference to erythropoietin	Erythropoietin, EPO	OR	
Reference to performance	Performance, endurance	OR	AND
Reference to subjects	Athletes	OR	

**Table 2 sports-13-00078-t002:** Physiotherapy evidence database (PEDro) ratings and Oxford levels of the included studies *.

Author (Year)	Items in the PEDro Scale													Evidence Level
	1	2	3	4	5	6	7	8	9	10	11	Total	Rating	
Andersen et al. (2023) [13]	Yes	1	1	1	1	1	0	1	1	1	1	9	Excellent	1b
Annaheim et al. (2016) [11]	Yes	1	1	1	1	1	0	1	1	1	1	9	Excellent	1b
Caillaud et al. (2015) [27]	Yes	1	0	1	0	0	0	1	1	1	1	6	Good	2b
Clark et al. (2017) [28]	Yes	1	1	1	1	1	0	1	1	1	1	9	Excellent	1b
Durussel et al. (2013) [10]	Yes	0	0	1	0	0	0	1	1	0	1	4	Fair	2b
Guadalupe-Grau et al. (2015) [29]	Yes	0	0	0	0	0	0	1	1	0	1	3	Poor	2b
Haider et al. (2020) [30]	Yes	1	1	1	1	1	0	1	1	1	1	9	Excellent	1b
Haile et al. (2019) [12]	Yes	0	0	1	0	0	0	1	1	1	1	5	Fair	2b
Heuberger et al. (2017) [31]	Yes	1	1	1	1	1	0	1	1	1	1	9	Excellent	1b
Sutehall et al. (2018) [32]	Yes	1	1	1	1	1	0	1	1	1	1	9	Excellent	1b
TOTAL												7.2	Good	6-1b/4-2b

* Items in the PEDro scale: 1 = eligibility criteria were specified; 2 = subjects were randomly allocated to groups (in a crossover study, subjects were randomly allocated an order in which treatments were received); 3 = allocation was concealed; 4 = the groups were similar at baseline regarding the most important prognostic indicators; 5 = there was blinding of all subjects; 6 = there was blinding of all therapists who administered therapy; 7 = there was blinding of all assessors who measured at least one key outcome; 8 = measures of at least one key outcome were obtained from more than 85% of the subjects initially allocated to groups; 9 = all subjects for whom outcome measures were available received the treatment or control condition as allocated or, where this was not the case, data for at least one key outcome was analyzed by “intention to treat”; 10 = the results of between-group statistical comparisons are reported for at least one key outcome; and 11 = the study provides both point measures and measures of variability for at least one key outcome. Scores: 1 = met; 0 = not met.

**Table 3 sports-13-00078-t003:** Characteristics of the included studies *.

Author (Year), Country	Sample Size (Sex), Groups	Age (Years), Participants	Intervention (Duration and Dose of rHuEPO)	Outcomes
Andersen et al. (2023) [13], Denmark	*n* = 48 (M/F) rHuEPO, *n* = 24 Placebo, *n* = 24	rHuEPO: 27.0 ± 6.0 Placebo: 26.0 ± 4.0 Recreational to well-trained athletes	Total duration: 6 weeks (2 weeks of baseline + 4 weeks of treatment), 3 weekly injections Dose: 9 IU·kg^−1^, placebo 0.9% NaCl Total dose: 108 IU·kg^−1^ All participants received oral doses of 80 mg ferrous sulfate	tHb: +6.7% Hct: +2.6% VO_2peak_: +4.2% P_peak_: +2.9% MPO: +4.1% t_lim_: +4.3%
Annaheim et al. (2016) [11], Switzerland	*n* = 40 (M) H, *n* = 10 MED, *n* = 10 L, *n* = 10 Placebo, *n* = 10	H: 35.7 ± 8.0 MED: 35.7 ± 5.4 L: 31.4 ± 7.7 Placebo: 31.2 ± 6.0 Endurance-trained athletes	Total duration: 4 weeks, injections every 2–3 days Dose: H 133 IU·kg^−1^, MED 66 IU·kg^−1^, and L 35 IU·kg^−1^, placebo 0.9% NaCl Total dose: H 1596 IU·kg^−1^, MED 792 IU·kg^−1^, and L 420 IU·kg^−1^ All participants received 100 mg of intravenous iron-III-saccharate	Hb: L +8.5%, MED +8.3%, and H +14% Hct: L +11.8%, MED +11.4%, and H +18.8% VO_2max_: L +2.6%, MED +5.7%, and H +5.6% t_lim_: L +16.7% (no sig.), MED +44.8%, and H +69.7% P_max_: L +1.9% (no sig.), MED +2.7% (sig. only with respect to baseline values), and H +5.8% RPE in P_max_: L unchanged, MED +3.5%, and H +5.4% (no sig. in all groups)
Caillaud et al. (2015) [27], France	*n* = 12 (M) rHuEPO, *n* = 6 Placebo, *n* = 6	rHuEPO: 26.8 ± 4.1 Placebo: 26.8 ± 6.0 Healthy aerobically trained athletes	Total duration: 4 weeks, injections every 3 days Dose: 50 IU·kg^−1^, placebo 0.9% NaCl Total dose: 450 IU·kg^−1^ All participants received adequate oral doses of ferrous sulfate, vitamin B_9_ and vitamin B_12_	Hb: +9.5% Hct: +12.3% VO2max: +6.3% MPO: +8.9% (no sig.) RER: −3.3% Carbohydrate oxidation rate: −18.3%
Clark et al. (2017) [28], Australia	*n* = 24 (M/F) H, *n* = 8 HP, *n* = 4 C, *n* = 8 CP, *n* = 4	H: 28.1 ± 4.6 HP: 34.2 ± 13.4 C: 36.3 ± 8.5 CP: 33.5 ± 11.4 Healthy recreational athletes	Total duration: 6 weeks Dose: H 3 weekly injections of 250 IU·kg^−1^ for 2 weeks + 4 weeks of wash-out period, C 3 weekly injections of 250 IU·kg^−1^ for 2 weeks + 3 weekly injections of 10 IU·kg^−1^ for 3 weeks, HP and CP same timing and NaCl dose Total dose: H 1500 IU·kg^−1^ and C 1590 IU·kg^−1^ All participants received 100 mg of intravenous iron in the first 2 weeks of treatment	tHb: H and C +18.4% (week 2), remains elevated at weeks 5 and 6, respectively Hct: H and C +7.9% (week 2), remains elevated at weeks 5 and 6, respectively VO_2peak_: H +6.2% and C +6.1% (week 2), remains elevated (+5.9%) in week 5 in group C TTE: H and C +28 s (week 2) and also increases by 28 s at week 5 in group C (both results not sig., but with a tendency to increase) [La^−^]: H and C −2.2 mmol·L^−1^ (week 2) and remains at −2.3 mmol·L^−1^ at week 6 in group H (sig. in both cases only with respect to baseline values)
Durussel et al. (2013) [10], UK	*n* = 19 (M) Runners, *n* = 10 Other activities, *n* = 9	Runners and other activities: 26.0 ± 4.5 Endurance-trained athletes	Total duration: 8 weeks Dose: 50 IU·kg^−1^ every 2 days for 4 weeks for all participants Total dose: 700 IU·kg^−1^ No treatment from week 5 to week 8 All participants received 100 mg of oral elemental iron daily during the 4 weeks of treatment	tHb: +19.7% (week 5) and +7.9% (week 8) VO_2max_: +8.4% (week 5) and +3.6% (week 8) Time in 3000 m: Runners: −5.2% (week 5) and −3.1% (week 8) Other activities: −6.3% (week 5) and −3.6% (week 8) Average: −5.7% (week 5) and −3.3% (week 8)
Guadalupe-Grau et al. (2015) [29], Denmark	*n* = 6 (M)	21.0 ± 2.0 Healthy athletes	Total duration: 8 weeks Dose: Injections of 5000 IU every day for one week + depending on Hct level 0, 2500 or 5000 IU once a week for 7 weeks Total dose: Varies All participants received 100 mg of oral iron daily during treatment, starting 2 weeks prior to treatment	Hb: +12.4% VO_2max_: +8.2% Fat_max_: +8.3% Mb: +9.4% CS activity: +7.8% Capillaries per area: +13.7%
Haider et al. (2020) [30], Switzerland	*n* = 29 (M/F) 4 groups: rHuEPO males and females (rHuEPO_m_ and rHuEPO_f_) and placebo males and females Cross-over study	25.0 ± 3.0 Healthy athletes	Total duration: 4 weeks Single injection of 60,000 IU (923 IU·kg^−1^) diluted in 250 mL of NaCl and placebo 250 mL of NaCl 24 h before tests After 4 weeks of wash-out period the subjects received the alternate treatment, and the next day repeated the tests performed on the first day in the same order	Hb: Placebo 142 g·L^−1^ vs. rHuEPO 144 g·L^−1^ Hct: Placebo 41.8% vs. rHuEPO 42.4% VO_2peak_: Placebo 45.1 mL·kg^−1^·min^−1^ vs. rHuEPO 46.1 mL·kg^−1^·min^−1^ P_peak_: Placebo 3.5 W·kg^−1^ vs. rHuEPO 3.5 W·kg^−1^ Total distance covered: Placebo 7.4 km vs. rHuEPO 7.4 km MPO: Placebo 175 W vs. rHuEPO 175 W Note: No significant differences between groups in all parameters analyzed
Haile et al. (2019) [12], UK	*n* = 39 (M) Kenyan, *n* = 20 Caucasian, *n* = 19	Kenyan: 26.4 ± 4.1 Caucasian: 26.0 ± 4.5 Endurance runners	Total duration: 8 weeks (4 weeks of treatment + 4 weeks of wash-out period), injections every 2 days Dose: 50 IU·kg^−1^ in both groups Total dose: 700 IU·kg^−1^ in both groups All participants received oral doses of 100 mg ferrous sulfate for all 4 weeks of treatment	Hb: Kenyan +10.1% (week 4) and +5.9% (sig. only with respect to baseline values) (week 8), Caucasian +16.2% (week 4) and +8.2% (week 8) Hct: Kenyan +12.1% (week 4) and +5.1% (week 8), Caucasian +18.6% (week 4) and +8.7% (week 8) VO_2max_: Kenyan +5.8% (week 4) and +2.6% (no sig.) (week 8), Caucasian +8.1% (week 4) and +4.4% (week 8) (Kenyan week 4 and Caucasian weeks 4 and 8, sig. only with respect to baseline values) Time in 3000 m: Kenyan −4.6% (week 4) and −3.3% (week 8), Caucasian −5.7% (week 4) and −3.4% (week 8) (all values analyzed sig. only with respect to baseline values)
Heuberger et al. (2017) [31], Netherlands	*n* = 48 (M) rHuEPO, *n* = 24 Placebo, *n* = 24	rHuEPO: 33.5 [22.0–48.0] Placebo: 33.5 [20.0–50.0] Healthy, well-trained but non-professional cyclists	Total duration: 8 weeks, injections once a week Dose: 5000 IU in the first 4 injections; if Hb was below target range, then dose increased to 6000 IU, 8000 IU or 10,000 IU; if in range, dose decreased to 2000 IU and if above, placebo was given (0.9% NaCl) Total dose: Varies All participants received oral doses of 200 mg ferrous fumarate and 50 mg ascorbic acid	Hb: +12% Hct: +16% VO_2max_: Placebo +3.8% and rHuEPO +10% P_max_: Placebo +1.8% and rHuEPO +5.4% Gross efficiency: Placebo −0.5% and rHuEPO −0.9% MPO: Placebo +4.6% and rHuEPO +5.4% Mont Ventoux climbing time: Placebo 1 h 40 min 15 s and rHuEPO 1 h 40 min 32 s Note: Significant differences between groups only exist in Hb, Hct, VO_2max_, and P_max_
Sutehall et al. (2018) [32], UK	*n* = 14 (M) rHuEPO, *n* = 7 Placebo, *n* = 7	rHuEPO and placebo: 30.0 ± 4.0 Endurance-trained athletes	Total duration: 9 weeks, injections 2 times a week Dose: 20 IU·kg^−1^ at weeks 3 and 9, 30 IU·kg^−1^ at weeks 4 and 8, and 40 IU·kg^−1^ at weeks 5, 6, and 7, placebo 0.9% NaCl Total dose: 440 IU·kg^−1^ All participants received daily oral doses of 105 mg of elemental iron or lactose tablets (placebo group) in the same amount	tHb: +11.3% (day 38 post first rHuEPO injection) and +2.3% (day 21 post last rHuEPO injection) VO_2max_: +3.9% VT: No significant results RSA: No significant results in anaerobic parameters, the only exception being the 7 sprint, where the time to reach maximum power was significantly lower in rHuEPO group (−0.67 s)

* Note: The results shown are statistically significant (sig.) and mainly pertain to the group that received rHuEPO administration, expressing as a result the change in percentage from the pre-treatment period to the end of treatment in each variable analyzed. The exposed variables are also significantly different with respect to the comparison groups at the end of treatment (mainly placebo group). In those cases where the results are not statistically significant but are necessary to make a comparison, they have been described as such (no sig.). Abbreviations: C = combined dose; CP = combined-placebo dose; CS = citrate synthase; F = female; Fat_max_ = exercise intensity where maximum fat oxidation occurs; H = high dose; Hb = hemoglobin; Hct = hematocrit; HP = high-placebo dose; IU = international units; L = low dose; [La^−^] = blood lactate concentration; M = male; Mb = myoglobin; MED = medium dose; MPO = mean power output; NaCl = sodium chloride; P_max_ = maximal power output; P_peak_ = peak power output; RER = respiratory exchange ratio; rHuEPO = recombinant human erythropoietin; RPE = rate of perceived exertion; RSA = repeated sprint ability; tHb = total hemoglobin mass; t_lim_ = constant load test limit time; TTE = time to exhaustion; VO_2max_ = maximal oxygen uptake; VO_2peak_ = peak oxygen uptake; VT = ventilatory threshold; and W = watt.

## Data Availability

No new data was created or analyzed in this study. Data sharing is not applicable to this study.

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
