# Peer review of "Recombinant Human Erythropoietin Effects on Well-Trained Athletes’ Endurance Performance: A Systematic Review"

_sports, 2025, doi:10.3390/sports13030078_

Round 1

Reviewer 1 Report

Comments and Suggestions for Authors

Thank you for the opportunity to review this manuscript. The Authors performed a systematic review of the effects of EPO on performance and power output parameters in well-trained athletes. The methodology is well-described, and the aim of the paper is well-defined. However, it is unclear to this Reviewer what further investigations the presented systematic review prompts, considering that EPO has been on the WADA list for 25 years, and there is no intention to have it removed from the prohibition list for its known performance-enhancing effects. What do the findings add to existing reviews? How could the elucidation of effects on submaximal performance-related parameters influence the perception of EPO usage in competitive sports? Is there an ethical way to conduct these studies?

Please find my specific comments below:

Line 30. "Expressed in international units..." reads as a fragmented sentence, please restructure.

It is unclear what Figure 1 refers to in the grey boxes "Identification of new studies via other methods". Please clarify.

Table 4 is a very busy one, and many abbreviations seem unjustified and unnecessarily add to the complexity, such as KEN = Kenyan athletes, CAU = caucasian athletes, PLA = placebo, BL = baseline etc.

Most of the Discussion section reads more like Results, while that section is confined to Table 4. Please revise and reorganize these two sections.

Line 254 states that the ergogenic effects are related to the challenge of detection. Please clarify.

Reviewer 2 Report

Comments and Suggestions for Authors

Thank you for the opportunity to comment on the systematic review by Spanish scientists. I found this topic interesting and I’ve got some ideas on how to improve their article. Please find detailes below. 

Major comments:

  • Why did you use such a limited number of keywords specified in the Table 1?
  • I think that the exact search code should be provided. in the supplementary material.
  • Why did you set the. cutoff for VO2 as in line 99? Do you have any reference to support that?
  • Did you search independently by two reviewers? How do you resolve any ambiguity in the inclusion of a particular study, or whether there were any such ambiguities? 

Minor comments:

  • specify in the line 13 the period of searching
  • Provide the manufacturer and version for Rayyan systems software. 
  • Table 2 is not necessary as it is extracted from another article. 
  • Remember that the key paper about the hematological reference values (in particular total hemoglobin mass) in elite endurance athletes should be discussed. See doi: 10.1080/02640414.2025.2453347.

To sum up my review, I recommend the authors to revise their paper properly and clarify my major comments. 

Round 2

Reviewer 1 Report

Comments and Suggestions for Authors

I greatly appreciate that the Authors carefully and fully addressed all my comments and have significantly improved the manuscript. The manuscript is relevant and interesting and contributes to the scene of sports medicine on a controversial topic. I have no further remarks.

Reviewer 2 Report

Comments and Suggestions for Authors

The authors revised their article properly.